# Effect of Hypoxia Preconditioned Secretomes on Lymphangiogenic and Angiogenic Sprouting: An in Vitro Analysis

**DOI:** 10.3390/biomedicines8090365

**Published:** 2020-09-20

**Authors:** Philipp Moog, Rahmin Schams, Alexander Schneidinger, Arndt F. Schilling, Hans-Günther Machens, Ektoras Hadjipanayi, Ulf Dornseifer

**Affiliations:** 1Experimental Plastic Surgery, Clinic for Plastic, Reconstructive and Hand Surgery, Klinikum Rechts der Isar, Technische Universität München, D-81675 Munich, Germany; philippmoog@web.de (P.M.); rahmin@gmx.de (R.S.); schneidinger@gmx.at (A.S.); e.hadjipanayi@googlemail.com (E.H.); ulf.dornseifer@isarklinikum.de (U.D.); 2Department of Plastic, Reconstructive and Aesthetic Surgery, Isar Klinikum, D-80331 Munich, Germany; 3Department of Trauma Surgery, Orthopedics and Plastic Surgery, Universitätsmedizin Göttingen, D-37075 Göttingen, Germany; arndt.schilling@med.uni-goettingen.de

**Keywords:** adipose-derived stem cells, adipose-derived cell supension, peripheral blood cells, blood-derived therapy, hypoxia, angiogenesis, hypoxia preconditioned plasma, hypoxia preconditioned serum, lymphangiogenesis, lymphatic regeneration

## Abstract

Hypoxia Preconditioned Plasma (HPP) and Serum (HPS) are two blood-derived autologous growth factor compositions that are being clinically employed as tools for promoting tissue regeneration, and have been extensively examined for their angiogenic activity. As yet, their ability to stimulate/support lymphangiogenesis remains unknown, although this is an important but often-neglected process in wound healing and tissue repair. Here we set out to characterize the potential of hypoxia preconditioned secretomes as promoters of angiogenic and lymphangiogenic sprouting in vitro. We first analysed HPP/HPS in terms of pro- (VEGF-C) and anti- (TSP-1, PF-4) angiogenic/lymphangiogenic growth factor concentration, before testing their ability to stimulate microvessel sprouting in the mouse aortic ring assay and lymphatic sprouting in the thoracic duct ring assay. The origin of lymphatic structures was validated with lymph-specific immunohistochemical staining (Anti-LYVE-1) and lymphatic vessel-associated protein (polydom) quantification in culture supernatants. HPP/HPS induced greater angiogenic and lymphatic sprouting compared to non-hypoxia preconditioned samples (normal plasma/serum), a response that was compatible with their higher VEGF-C concentration. These findings demonstrate that hypoxia preconditioned blood-derived secretomes have the ability to not only support sprouting angiogenesis, but also lymphangiogenesis, which underlines their multimodal regenerative potential.

## 1. Introduction 

Complete restoration of physiological tissue architecture is the ultimate goal of regenerative medicine [1]. In humans, restorative responses following dramatic injuries or chronic wounds are limited, generally leading to poor wound healing with scarring, rather than full regeneration of traumatized body parts [1]. Considering the fact that wounds naturally heal via a set of complex and interactive processes, including hemostasis, inflammation, proliferation, and remodeling [2,3,4], gives an idea of the complexity of tissue repair. Research on regenerative treatments commonly focuses on stimulating angiogenesis (formation of new blood vessels) and improving tissue perfusion, in order to provide an adequate supply of oxygen/nutrients to the wound bed, which forms an absolute prerequisite for optimal cellular functions. It is gradually becoming apparent, however, that modern regenerative strategies must employ a multimodal approach that reaches beyond angiogenesis, also targeting other important processes involved in reparation of injured tissue.

Unlike angiogenesis, lymphangiogenesis is often neglected in the context of tissue regeneration, and has been so far poorly studied. Nonetheless, the importance of a sufficient lymphatic network for optimal tissue metabolism and an intact immune response must not be underestimated. The main function of the lymphatic vasculature is interstitial fluid drainage [5], which is important for fluid homeostasis, immune cell surveillance and trafficking, and lipid absorption [5,6,7,8,9,10]. In inflammatory settings, lymphangiogenesis facilitates the resolution of tissue edema and promotes macrophage and dendritic cell mobilization [11,12,13,14]. If the local immune response is delayed, or interstitial edema reduces cellular gas exchange and nutrition (by increasing diffusion distance), the wound healing process is ultimately impaired [6,15,16].

Despite the lack of an adequate body of research work, it is becoming increasingly evident that the complex cell-growth factor interactions, which are known to play catalytic roles in angiogenesis, appear to be involved in lymphangiogenesis as well [17]. For example, the VEGF family of protein factors, i.e., VEGF-A, -C, and -D can induce the outgrowth of lymphatic vessels [18], while VEGF-C/VEGFR-3 signaling is a primary mediator of lymphangiogenesis [11,19]. Moreover, the angiogenesis-related growth factors fibroblast growth factor 2 (FGF-2), insulin-like growth factor 1 (IGF-1), IGF-2, hepatocyte growth factor (HGF), endothelin-1 (ET-1), and PDGF-B have all been reported to induce lymphangiogenesis in different contexts [5,20]. Some studies have also pointed to the regulatory effects of TSP-1 and PF-4 as key endogenous inhibitors of lymphangiogenesis, through the inhibition of lymphatic endothelial cell migration and proliferation [21,22].

Our previous work has provided strong evidence that hypoxia preconditioned blood-derived secretomes could constitute a new generation of autologous and bioactive injectable/topical compositions that can supply the necessary biochemical signals for stimulating angiogenesis and driving wound healing to completion [23,24,25,26,27,28]. These angiogenic compositions can be obtained through the method of Extracorporeal Wound Simulation (EWS), using peripheral (venous) blood [23,24,25,26,27,28,29]. The ability to condition PBCs under the same environmental conditions encountered within a wound, i.e., physiological temperature and hypoxia, offers an easy way to optimize the angiogenic potential of Hypoxia Preconditioned Plasma (HPP) and Hypoxia Preconditioned Serum (HPS), which can be differentially prepared by adjusting blood coagulation prior to hypoxic conditioning [23,24,25,26,27,28,29,30]. More specifically, we have shown that the angiogenic potential of blood-derived secretomes is defined by the complex stoichiometry of their component pro- and anti-angiogenic factor proteins, rather than the concentration of one or more individual growth factors [28,29]. These secretomes have also been extensively examined for their angiogenic activity, showing that they are able to induce microvessel formation and sprouting in vitro, as well as promote wound healing in vivo [23,27,28]. The angiogenic potency of hypoxia preconditioned secretomes is further highlighted by the fact that they maintain their pro-angiogenic activity in vitro, even when they are prepared from peripheral blood that is obtained from patients who receive oral anticoagulation due to underlying vascular pathology or who suffer from diabetes mellitus [30].

To our knowledge, the lymphangiogenic potential of hypoxia preconditoned blood-derived secretomes remains unknown. Provided that there seems to be a significant overlap between the cellular/growth factor processes and regulatory pathways involved in angiogenesis and lymphangiogenesis, it is indeed possible that these secretomes might also have the capacity to stimulate the formation of lymphatic vessels. In the current study, we aimed to characterize the proteomic composition of hypoxia preconditioned plasma (HPP) and serum (HPS), in terms of key pro- and anti-angiogenic protein factors that may be involved in lymphangiogenesis (VEGF, PF-4, TSP-1), and analyze their ability to promote angiogenic and lymphangiogenic sprouting in vitro. Beyond providing useful insights into the clinical utility of these therapeutic blood-derived products, this work could also enhance our scientific understanding on the biological processes that physiologically link angiogenesis with lymphangiogenesis.

## 2. Experimental Setup

### 2.1. Ethical Approval

All blood donors provided written informed consent as directed by the ethics committee of the Technical University Munich, Germany, which approved this study (File Nr.: 497/16S; Amendment, date of approval: 11 November 2016).

### 2.2. Preparation of Blood Plasma/Serum and Hypoxia Preconditioned Plasma (HPP)/Serum (HPS) Samples

Peripheral venous blood (20 mL) was collected from young healthy non-smoking subjects (n = 5), who were not taking any medication and without known comorbidities, in a 30 mL polypropylene syringe (Omnifix^®^, Braun AG, Melsungen, Germany) that contained no additive for normal serum and HPS preparation or was prefilled with 1-mL heparin (Medunasal^®^, Heparin 500 I.U. 5-mL ampoules, Sintetica^®^, Münster, Germany) for normal plasma and HPP preparation, under sterile and standardized conditions (Blood Collection Set 0.8 × 19 mm × 178 mm; Safety-Lok, CE 0050, BD Vacutainer, BD, Franklin Lakes, NJ, USA). For normal plasma/serum preparation, following passive sedimentation for 60 min at room temperature (25 °C, no centrifugation) the blood was separated into three layers, from bottom to top: red blood cell component (RBCs), clot/buffy coat, and serum/plasma, so that the top layer (serum or plasma) could be filtered into a new syringe. For HPP/HPS preparation, following blood sampling a 0.2-µm pore filter was attached to the syringe (Sterifix^®^, CE 0123, Braun AG, Melsungen, Germany), before adding 5 mL of air into the syringe through the filter, by withdrawing the plunger. Subsequently, the filter was removed, and the capped syringe was placed upright in an incubator (37 °C/5% CO_2_). Incubation was carried out for 4 days (blood incubation time), without any prior centrifugation. Pericellular (local) hypoxia (~1% O_2_) was generated in situ through cell-mediated O_2_ consumption, by controlling the blood volume per unit area (BVUA > 1 mL/cm^2^) in the blood-containing syringe, and consequently, the PBC seeding density [25,29,30]. After 4 days, the blood was passively separated into three layers, from bottom to top: red blood cell (RBC) component, buffy coat/clot, and HPP/HPS, so that the top layer comprising hypoxia preconditioned plasma or serum could be filtered (0.2-µm pore filter, Sterifix^®^, Braun AG, Melsungen, Germany) into a new syringe, removing cells/cellular debris. 

### 2.3. Preparation of Adipose-Derived Cell Suspension (ADCS) 

ADCS is known for its pro-angiogenic and lymphangiogenic activity [31,32,33,34], therefore it was selected as a positive control for this study. According to manufacturer’s (ARC Processing System, InGeneron Inc., Houston, TX, USA) instructions and a previously published protocol [35], adipose tissue was dissected from female adult mice, (approximately 10–12 g) using appropriate surgical techniques into a sterile container and then manually minced until average piece size was approximately 2 mm or less. The minced adipose tissue was transferred into a sterile tube and lactated ringer (preheated to 39 °C) was added to a fill level of 30 mL. Then 2.5 mL of the reconsitituted Matrase reagent (ARC System, In Generon Inc., Houston, TX, USA) was added to the tube containing the adipose tissue. The tube was inverted several times to mix thoroughly and placed in an inverted position into the processing system for repetitive acceleration and deceleration for 30 min at 39 °C. Processing resulted in three layers: a thin layer of oil on the top, a milky adipose layer in the middle, and an aqueous layer on the bottom. Then the tube assembly was inverted, so that the tube was on the top and the filter (Steriflip Filter) (ARC Processing System, InGeneron Inc., Houston, TX, USA) at the bottom, for allowing separation of the different layers for 1 min. The filtered material was then centrifuged again (600× *g* for 5 min). After centrifugation, a pellet of red and white colored cells formed at the bottom with distinct aqueous, adipose, and oil layers settling above. Supernatants were decanted (aqueous, adipose, and oil layers) into a sterile waste container. Then 40 mL lactated ringer was added to the cell pellet and centrifuged again (600× *g* for 5 min). The decanting and washing of the cell pellet, in a manner as described above was repeated to get adipose-derived cells. Then lactated ringer solution was added, using the 3 mL syringe (ARC Processing System, InGeneron Inc., Houston, TX, USA) and a new 18 G needle, very gently to the cell pellet at the bottom of the tube. The final cell product was drawn slowly in and out of the 3 mL syringe (ARC Processing System, InGeneron Inc., Houston, TX, USA), three times to break up cell clumps. After this final isolation step, adipose-derived cells (ADCs) were resuspended in Dulbecco’s Modified Eagle’s Medium (D-MEM without serum, Life Technologies, Paisley, UK) to reach a cell concentration of 3500 cells per mL. Cell counts were determined using the CASY Cell Counter & Analyzer (OLS OMNI Life Science GmbH & Co KG, Bremen, Germany), as described by the manufacturer’s protocol. Then adipose-derived cell suspension (ADCS) was ready for testing, without any further conditioning.

### 2.4. Quantitative Analysis of VEGF-C, TSP-1, PF-4 Concentration in Blood-Derived Secretomes

For analysis of protein factor concentration, blood-derived secretomes (hypoxia preconditioned plasma (HPP) and serum (HPS), normal plasma and serum), were sampled following 4 days incubation and analyzed by ELISA for VEGF-C, TSP-1, PF-4 (R&D Systems, Inc., Minneapolis, MN, USA), according to manufacturer’s instructions. Five samples were tested per condition.

### 2.5. Analysis of the Effect of Blood-Derived Secretomes on Microvessel Sprouting In Vitro

Blood-derived secretomes (hypoxia preconditioned plasma (HPP) and serum (HPS), normal plasma and serum), were tested in the mouse aortic ring assay, in order to assess their ability to induce microvessel sprouting. Aortic rings were dissected from female adult mice as previously described [36], underwent overnight serum starvation in opti-MEM reduced serum medium (Life Technologies, Darmstadt, Germany), before being embedded into Matrigel bilayer matrix (50 μL/layer in 96-well plates) (BD, Heidelberg, Germany). Secretomes and control media samples (VEGF 90 ng/ mL, ADCS; positive controls/phosphate buffered saline (PBS); negative control) samples were added (150 μL/well) to the rings, before culturing them in a 5% CO_2_/37 °C incubator. Medium change was carried out every 3 days, while rings were observed with phase contrast microscopy at 0, 3, 6 and 8 days and photographed, with all four quarters per ring analyzed for sprouting (sprouting was defined as the formation of structures of connected cells that were attached, at their base, to the ring). Furthermore, tube length was quantified after a culture period of 8 days using image analysis using ImageJ software (NIH, Bethesda, MD, USA). At least three aortic rings were tested per condition. 

### 2.6. Analysis of the Effect of Blood-Derived Secretomes on Lymphatic Vessel Sprouting In Vitro

To assess the ability of blood-derived secretomes (hypoxia preconditioned plasma (HPP) and serum (HPS), normal plasma and serum) to induce lymphatic vessel sprouting, these were tested in the thoracic duct ring assay. Thoracic duct rings were dissected from female adult mice (Figure 1) as previously described [37], underwent overnight serum starvation in opti-MEM reduced serum medium (Life Technologies, Darmstadt, Germany), before being embedded into Matrigel bilayer matrix (50 μL/layer in 96-well plates) (BD, Heidelberg, Germany). Secretomes and control media samples (heat-decomplemented fetal calf serum (FCS 20%) (GIBCO FCS, Thermo Fisher, Wien, Austria); ADCS; positive controls/phosphate buffered saline (PBS); negative control) were added (150 μL/well) to the rings, before culturing them in a 5% CO_2_/37 °C incubator. Medium change was carried out every 2 days, while rings were observed with phase contrast microscopy at 0, 2, 4, 6 and 8 days and photographed, with all 4 quarters per ring analyzed for sprouting (sprouting was defined as the formation of structures of connected cells that were attached, at their base, to the ring). Furthermore, tube length was quantified after a culture period of 8 days using image analysis with ImageJ software (NIH, Bethesda, MD, USA). At least three thoracic duct rings were tested per condition.

### 2.7. Immunohistochemical Staining of Lymphatic Sprouts

Lymphatic sprouts were stained with anti-lymphatic vessel endothelial hyaluronan receptor-1 (Anti-LYVE-1-antibody) (Abcam, Cambridge, MA, USA), a specific lymphatic endothelial cell marker, according to the manufacturer’s instructions. The staining was supplemented with nuclear DAPI counterstain (Abcam; Cambridge, MA, USA), according to the manufacturer’s instructions. At least three thoracic duct rings were tested per condition.

### 2.8. Quantitative Analysis of Polydom Concentration in Thoracic Duct Ring Assay

Thoracic duct ring assay culture supernatants were sampled following medium change at 2 and 8 days and analyzed by ELISA for polydom/svep1, a cell-adhesion receptor involved in lymphangiogenesis [38,39], according to the manufacturer’s (MyBioSource; San Diego, CA, USA) instructions. Three samples were tested per condition.

### 2.9. Statistical Analysis

Five subjects were tested for each experimental condition. Data are expressed as mean ± standard deviation, as noted. Statistical analysis was carried out using Student’s independent t-test, when a maximum of two groups was compared, or one-way ANOVA with Bonferroni adjustment, accompanied by post-hoc pairwise comparisons for analysis of more than two groups, using SPSS 14 software (version 14, IBM, Ehningen, Germany). The probability of a type one error was set to 5% (α = 0.05), unless otherwise noted.

## 3. Results

### 3.1. Quantitative Analysis of pro- (VEGF-C) and anti- (TSP-1, PF-4) Angiogenic/Lymphangiogenic Growth Factor Concentration in Hypoxia Preconditioned Blood-Derived Secretomes

To establish a growth factor concentration baseline, we first quantitatively analyzed via ELISA the concentration of key angiogenesis-related protein factors in normal plasma and serum, and compared them to their hypoxia-conditioned counterparts. As shown in Figure 2, the concentration of VEGF-C in hypoxia preconditioned plasma (HPP) and hypoxia preconditioned serum (HPS) showed a 3- to 5-fold increase compared to baseline level in fresh plasma (*p* = 0.19) and fresh serum (*p* = 0.0459), respectively. No significant difference was observed between HPP and HPS VEGF-C level (*p* = 0.3). The concentration of the platelet-derived angiogenic inhibitors TSP-1 and PF-4 in fresh plasma and HPP was significantly lower than that in HPS (*p* < 0.05), indicating that the process of blood conditioning used for HPP preparation did not promote significant platelet activation. Furthermore, the concentration of TSP-1, but not PF-4, in HPS was significantly higher compared to that in fresh serum (*p* < 0.05).

### 3.2. Analysis of the Ability of Hypoxia Preconditioned Blood-Derived Secretomes to induce Angiogenic Sprouting In Vitro

Having assessed the growth factor composition of hypoxia preconditioned blood-derived secretomes, in terms of key pro- and anti-angiogenic protein factors, an analysis of microvessel sprouting was carried out using the mouse aortic ring assay. In all secretome cultures, we observed a trend for an increasing number of sprouts as culture duration increased from 3 to 6 and 8 days (Figure 3A,B). Hypoxic preconditioning of blood-derived secretomes resulted in a higher sprout number and length, which could initially be seen as a significant difference between HPP and fresh plasma cultures at 3 days, with the difference persisting up to 8 days (Figure 3A–C) (*p* < 0.05). Similarly, HPS appeared to generate more vascular sprouts than fresh serum, at all time points examined, although such differences were not statistically significant due to a high standard deviation between samples. The pro-angiogenic effect of hypoxic conditioning was also evident in the greater number of sprouts induced by HPP compared to pure recombinant VEGF (pos. control). Specifically, there were approx. 5 times more sprouts in HPP cultures than in VEGF cultures on day 8 (*p* < 0.05). Due to the known ability of adipose-derived cell suspension (ADCS) to induce both angiogenic and lymphatic sprouting in vitro and in vivo [31,32,40,41], this was also investigated in addition to VEGF, as positive control in sprouting assays. ADCS appeared to have a similar activity as VEGF, and it was found to induce 3–5 times less sprouts than hypoxia preconditioned secretomes (Figure 3A,B). This difference was particularly evident in day 3 and day 8 cultures, where it was significant compared to HPP cultures (*p* < 0.05).

### 3.3. Qualitative and Quantitative Validation of the Lymphatic Sprouting Assay

Lymphatic sprouting was tested in the thoracic duct ring assay. Due to the technical difficulty associated with thoracic duct preparation, we first wanted to confirm the lymphatic origin of sprouts in the thoracic duct ring assay, through lymphoid-specific immunohistochemical staining and by lymphoid-specific protein (polydom) quantification, before proceeding to a quantitative assessment of lymphatic sprouting in blood-derived secretome cultures. 

As a first method for validating the lymphatic origin of sprouts observed in the thoracic duct ring assay, we examined sprouts for the presence of LYVE-1 (lymphatic vessel endothelial hyaluronan receptor-1), one of the most specific and widely used lymphatic endothelial markers that is located in lymph nodes and in the luminal/adluminal surfaces of lymphatic vessels [14,40]. Specific immunohistochemical staining (Anti-LYVE-1) of sprouts indicated lymphatic structures in HPP/HPS cultures and positive control (ADCS, FCS) cultures, while minimal to no lymphatic sprouting was observed in fresh plasma/serum cultures, as well as negative control (PBS) cultures (Figure 4A). In all culture conditions, thoracic duct rings stained positively (green straining) for LYVE-1, confirming their true lymphoid origin.

Polydom (also called Svep 1), a large extracellular matrix protein (>300 kDa) comprising Sushi, von Willebrand factor type A, EGF and pentraxin domain-containing protein 1, is a high-affinity ligand for integrin α9β1, a cell adhesion receptor involved in lymphangiogenesis [38,39]. We hypothesized that polydom protein would passively be released into the culture medium, provided that true lymphoid tissue was cultivated. The polydom concentration of culture supernatants was determined by ELISA on day 2 and day 8, following medium change. As can be seen in Figure 4B, polydom was detected in all culture supernatants, while there were no significant differences between 2 and 8 days of culture for any of the blood-derived secretomes tested, suggesting an early and steady release. HPP/HPS culture supernatants appeared to have almost half the polydom concentration of fresh plasma/serum and positive/negative control cultures. This difference was indeed significant for 8 days HPS cultures (*p* < 0.05). This data, together with the fact that polydom was also present in negative control (PBS) cultures, confirmed the lymphatic origin of the observed sprouts, while also suggesting that polydom leakage into the assay supernatant was more closely correlated with the presence of thoracic duct rings rather than with the degree of lymphatic sprouting.

### 3.4. Analysis of the Ability of Hypoxia Preconditioned Blood-Derived Secretomes to induce Lymphatic Sprouting In Vitro

Following this double validation method, a quantitative evaluation of the ability of blood-derived secretomes to induce lymphatic sprouting in thoracic duct ring assay in vitro cultures was carried out. Sprouts were analyzed using light microscopy. None of the secretomes tested appeared to induce sprouting on day 2 (Figure 5A,B). While lymphatic sprouting was first observed at 6 days in HPP cultures, HPS generated sprouts after only 4 days (Figure 5B). Importantly, at both these time points, HPP- and HPS-induced sprouting was significantly greater than what was observed in fresh plasma and serum cultures, respectively (*p* < 0.05). Interestingly, while no sprouting was observed in fresh plasma cultures, at any time point, a small number of sprouts could be seen in day 8 serum cultures, further pointing to the higher effectiveness of serum, compared to plasma, in stimulating lymphatic sprouting in vitro. While HPP showed steady sprouting from 6 to 8 days, HPS cultures saw a decline, reaching one third of the day 4 peak after 8 days, a level comparable to HPP cultures (Figure 5B).

In this experiment we included adipose-derived cell suspension (ADCS), a secretome with lymphangiogenic properties [34], and fetal calf serum (FCS), a powerful lymphangiogenic stimulator [10], as positive controls. ADCS stimulated steady sprouting from the 4th culture day onwards, while sprout numbers were always higher than those observed in negative control (PBS) cultures (*p* < 0.05) (Figure 5A,B). In contrast to HPP cultures, which elicited comparable sprouting as ADCS cultures at day 6 and day 8, HPS appeared to outperform ADCS at these time points, although differences were not significant due to a large standard deviation present between HPS samples. This relative difference seen between HPP and HPS cultures was also evident when comparing them with FCS, which itself appeared to induce a stronger response compared to ADCS (Figure 5B). Indeed, while HPS was found to be equally potent as FCS at 6 and 8 culture days, HPP promoted significantly less sprouts than FCS on day 6 (*p* < 0.05) (Figure 5B).

Examination of lymphatic sprout length at 8 days culture revealed that HPP- and HPS-induced sprouts had a comparable length as those induced by fresh serum, ADCS and FCS, and significantly greater length than sprouts generated in fresh plasma cultures (*p* < 0.05) (Figure 5C), suggesting that the initiation of lymphatic sprouting and the degree of sprout extension are differentially-regulated processes.

## 4. Discussion

Hypoxia preconditioned blood-derived secretomes represent a new generation of autologous growth factor preparations [23,24,25,26,27,28,30] that can be produced through extracorporeal conditioning of peripheral blood cells (PBCs) under wound-simulating conditions, namely physiological temperature and hypoxia [25,26]. We had previously demonstrated that hypoxia preconditioned plasma (HPP) and serum (HPS) supply angiogenesis-specific signaling, similar to that naturally produced within the wound microenvironment [23,28]. HPP and HPS organically differ with respect to their protein factor composition, since they correlate with distinct wound healing phases, the former having a direct correlation with the hypoxia-induced, angiogenesis-driven proliferative phase, while the latter also incorporating the platelet-derived heamostatic phase [25,27,28]. The clinical utility of these secretomes harnesses their angiogenic activity, despite their differences, since they can both provide a useful tool for stimulating microvessel sprouting and new vessel formation on demand [23,27,28]. As such, they could play an important role in a modern therapeutic strategy that aims to improve local tissue perfusion and accelerate wound healing where this is delayed or stagnant. 

The main thesis of this work is based on the notion, however, that regenerative wound therapy means more than just stimulating angiogenesis. Although angiogenesis appears to be the key driver of tissue repair and regeneration, lymphangiogenesis is of equal importance. Without sufficient lymphatic drainage of the interstitium, perivascular edema progressively impedes nutrient and oxygen supply, leading to cellular damage. Furthermore, lymphatic regeneration plays an important role in the inflammatory process of wound healing, by assisting the removal of local debris and inflammatory cells. Given the fact that angiogenesis and lymphangiogenesis are two complementary processes, compositions that have been shown capable of stimulating angiogenesis, such as hypoxia preconditioned blood-secretomes [23,24,25,26,27,28], might also support the formation of new lymphatic vessels. The findings of this study appear to support this hypothesis.

Quantitative analysis of angiogenic/lymphangiogenic growth factor concentration in blood-derived secretomes showed an increased concentration of VEGF-C in hypoxia preconditioned plasma (HPP) and hypoxia preconditioned serum (HPS) compared to their baseline level in fresh plasma and fresh serum, respectively (Figure 2A). It is known that, similar to blood vessel formation, the growth of lymphatic vessels is primarily mediated by vascular endothelial growth factor VEGF-C/VEGFR-3 signaling [11,19]. The current data verifies the positive effect of hypoxic conditioning on optimizing the pro-angiogenic/lymphangiogenic composition of blood-derived secretomes, shown here by the significant increase in HPS VEGF level compared to normal serum, and the significantly lower concentration of platelet-derived angiogenic inhibitors TSP-1 and PF-4 in HPP compared to HPS (Figure 2). While it has been reported that angiogenic inhibitors may have some elective effects on the formation of blood vessels, it remains unclear whether they also have additional inhibitory effects on lymphangiogenesis [42]. Our data showed a greater concentration of TSP-1 in HPS, compared to normal serum, suggesting that hypoxia-induced upregulation of TSP-1 was responsible for reaching a level beyond what is normally achieved through platelet activation. Various studies have demonstrated that TSP-1 acts as an inhibitor of inflammatory lymphangiogenesis in vitro *and vivo* [18,21]. For example, exposure of macrophages to TSP-1 suppresses the expression of lymphangiogenic factors (VEGF-C and VEGF-D), while the absence of TSP-1 leads to a significant increase in lymphangiogenesis in a model of inflammation [18]. The identification of TSP-1 as an endogenous inhibitor of lymphangiogenesis opens new avenues for understanding the complex interaction between this process and angiogenesis. Of equal importance, we found a significantly lower PF-4 concentration in HPP compared to HPS (Figure 2C). The literature suggests that PF-4 is able to dose-dependent inhibit the migration and proliferation of lymphatic endothelial cells [22]. Further work is required before clarifying whether the relative advantage of HPS, in having a higher VEGF concentration than HPP, is partly or even fully offset by its higher concentration of inhibitory factors, such as TSP-1 and PF-4, when considering the secretomes’ bioactivity in the context of lymphangiogenesis. 

Hypoxia preconditioned blood-derived secretomes (HPP, HPS) appeared to promote stronger microvessel sprouting in the aortic ring assay compared to normal plasma and serum (Figure 3A). The pro-angiogenic effect of hypoxic conditioning, also evident in a significantly higher number of HPP-induced sprouts compared to ADCS and pure VEGF (pos. controls) (Figure 3B), was in line with the ELISA results. There were no significant differences in sprout number or length between HPP and HPS (Figure 3B,C), suggesting that while pro-angiogenic factors, such as VEGF, are important for angiogenic sprouting, this process is not directly inhibited/limited by the presence of anti-angiogenic factors, e.g., TSP-1, PF-4, in agreement with our previous findings [28,29]. 

In order to more precisely quantify the lymphatic sprouting response in cultured thoracic duct rings, that were obtained through a technically demanding microsurgical preparation, the lymphatic origin of capillary-like structures observed to emerge from rings was first assessed with Anti-LYVE-1 immunohistochemical staining (Figure 4A), before quantifying the lymph-specific polydom concentration in culture supernatants (Figure 4B). Polydom is a large extracellular matrix protein of >300 kDa expressed in cultured bone marrow stromal cells and a high-affinity ligand for integrin α9β1, a cell adhesion receptor involved in lymphangiogenesis [38], which affects remodeling of lymphatic vessels in both mouse and zebrafish [38]. However, its physiological function remains unclear [38,39]. Here, a lower polydom concentration was recorded in HPP/HPS culture supernatants, a finding that is inherently difficult to explain. Provided that these secretomes promoted stronger lymphatic sprouting compared to normal plasma/serum and PBS negative control, as it was directly evident by immunohistochemical analysis (Figure 4A), it is indeed possible that in these cultures there was less unbound polydom protein available for leaking into the medium, which further implies that polydom may be involved in lymphatic sprouting. Further studies are required, however, before this basic hypothesis can be properly tested, which will also aid the understanding of the exact function of polydom in lymphangiogenesis.

Following this double validation method, we analyzed the ability of hypoxia preconditioned blood-derived secretomes to induce lymphatic sprouting in the thoracic duct ring assay. Like angiogenic sprouting, lymphatic sprouting is a complex process that depends on multiple cellular events, including cellular outgrowth from a preexisting lymphatic vessel, endothelial cell proliferation, migration and differentiation into capillary structures [37]. The thoracic duct ring assay forms an adaptation of the aortic ring assay, that has proved to be a useful tool for investigating sprouting angiogenesis [36,37,43], and was therefore also applied here. Our hypothesis that hypoxia preconditioned secretomes support lymphatic sprouting could be confirmed by the finding that both HPP and HPS induced more sprouting than non-hypoxia preconditioned secretomes (normal plasma and serum), as well as negative control PBS samples (Figure 5A,B). Importantly, the HPS-induced response was not weaker than that elicited by the lymphangiogenic stimulators FCS and ADCS [31,37], used here as positive controls (Figure 3B). In contrast, there appeared to be a delay in the HPP-induced response, which could first be observed on day 6 (i.e., later than 4 days, as for HPS), while the number of lymphatic sprouts was significantly lower than in FCS cultures (Figure 5B). It can therefore be concluded that HPS has a greater lymphatic sprouting activity than HPP, at least in vitro (note: it is unlikely that any difference seen was due to residual heparin in HPP, since we have previously shown that heparin does not either promote or inhibit angiogenic sprouting [28]). When attempting to understand why this might be the case, it is important to consider that lymphatic regeneration most likely begins/takes place during the inflammatory phase of wound healing (as also highlighted by the day 4 induction of sprouting in HPS cultures), and therefore likely requires platelet-derived signaling. This suggests that the presence of inhibitory factors, such as TSP-1 and PF-4, in HPS is not only counterproductive towards lymphatic sprouting, but very likely a key pre-requisite for an optimal response. Indeed, we could previously show that blocking PF-4 activity in the wound microenvironment using anti-PF-4 appeared to inhibit, rather than enhance angiogenic sprouting [29], which could mechanistically be explained through the regulation of matrix remodelling by MMPs expressed in vascular endothelial cells [44]. It appears then that PF-4 may exert an inhibitory effect on angiogenic sprouting at higher concentrations, but a minimum amount of this factor is nonetheless necessary for initiation of the sprouting process [29]. Our current findings point to the idea that this might also hold true for lymphatic sprouting. Admittedly, further work is necessary for better understanding the role of TPS-1 and PF-4 in lymphangiogenesis.

## 5. Conclusion 

The data presented here provide supporting evidence to our previous findings, and offer further insight into the clinical utility of hypoxia preconditioned blood-derived secretomes as a tool for stimulating lymphangiogenesis. Our results suggest that HPS may be a better alternative to HPP when choosing an autologous growth factor composition to promote lymphatic sprouting, and by extension lymphatic vessel regeneration. Importantly, this work contributes to the idea that angiogenesis and lymphangiogenesis have symbiotic roles in wound healing and tissue repair, since they both appear to rely on overlapping growth factor mechanisms. 

## Figures and Tables

**Figure 1 biomedicines-08-00365-f001:**
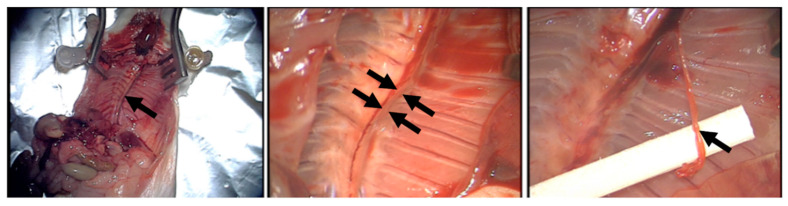
Thoracic Duct Ring Assay. Surgical removal of the thoracic duct, the main lymphatic vessel, under microscopic magnification. The aorta has already been excised and prepared for use in the aortic ring assay. Black arrow(s) show(s) thoracic duct.

**Figure 2 biomedicines-08-00365-f002:**
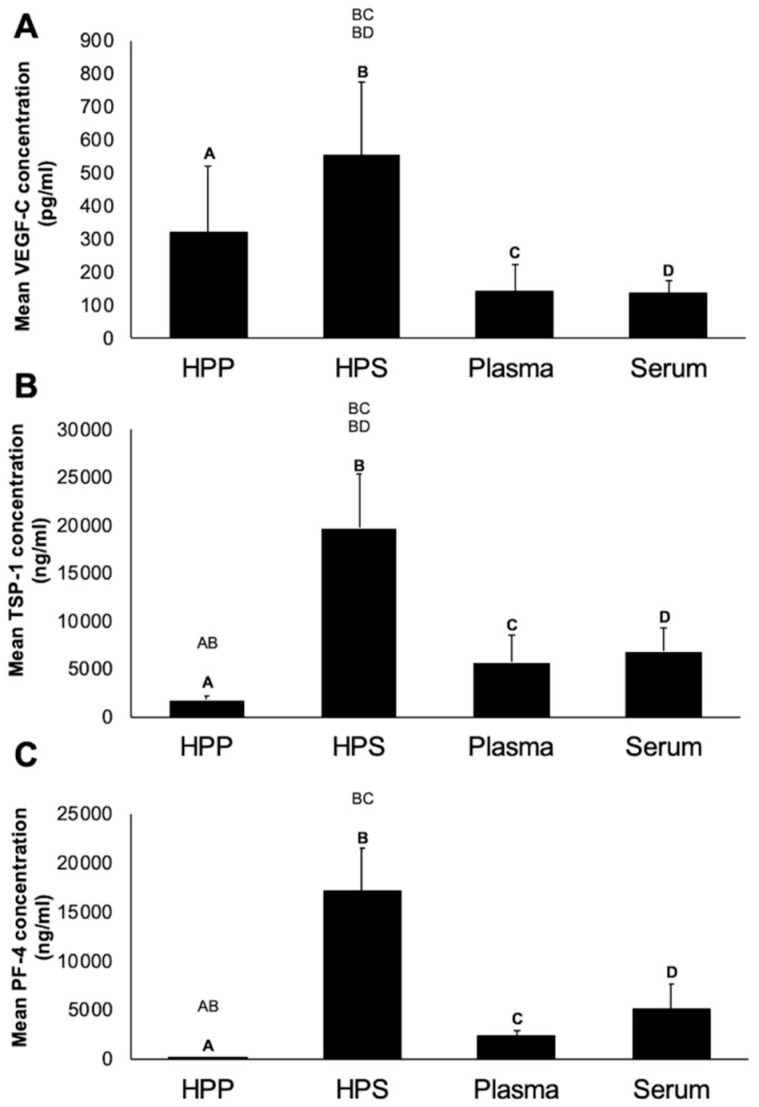
Quantitative analysis of pro- (VEGF-C) and anti- (TSP-1, PF-4) angiogenic/lymphangiogenic growth factor concentration in blood-derived secretomes. Plots showing the concentration of (**A**) VEGF-C (pg/mL), (**B**) TSP-1 (ng/mL), (**C**) PF-4 (ng/mL) in hypoxia preconditioned plasma (HPP) and serum (HPS), as well as fresh plasma and serum obtained from 5 young, healthy subjects as analysed by ELISA. Capital letter pairs over plots indicate statistical comparison of corresponding data points. For all pair comparisons, *p* < 0.05, unless otherwise indicated. Error bars represent s.d. (*n* = 5).

**Figure 3 biomedicines-08-00365-f003:**
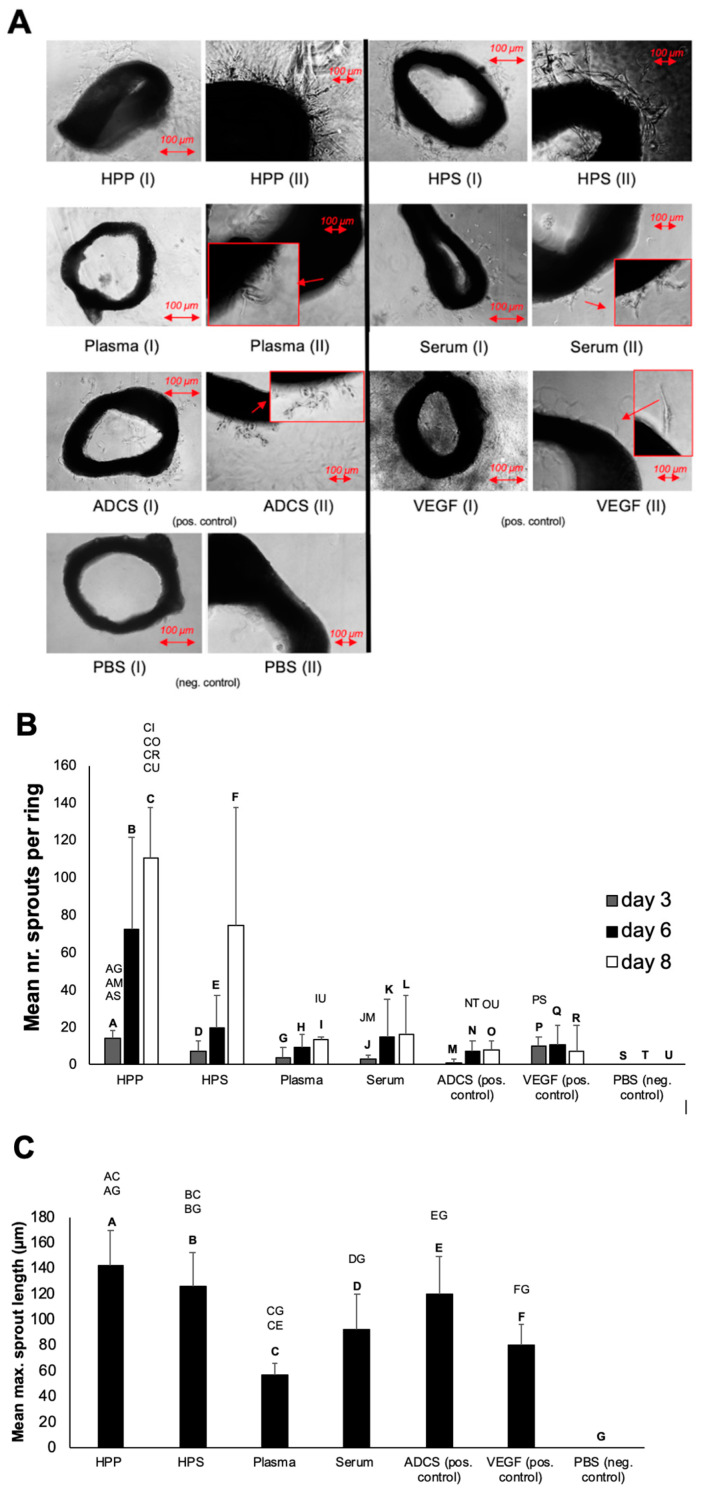
Effect of blood-derived secretomes on microvessel sprouting in the aortic ring assay. (**A**) Panel showing representative images of angiogenic sprouting assays (8 days culture), carried out in the presence of hypoxia preconditioned blood-derived secretomes (hypoxia preconditioned plasma (HPP) and serum (HPS)), normal plasma and serum, as well as positive controls (adipose-derived cell suspension (ADCS) and vascular endothelial growth factor (VEGF)) and negative control (phosphate buffered saline (PBS)) samples. Photographs were taken using a 10× (I) and a 20× (II) magnification. Enlarged image sections are indicated by red insets (Bars = 100 µm). (**B**) Plot showing the mean number of sprouts per ring over a culture duration of 3, 6 and 8 days (*n* = 3). (**C**) Plot showing the mean maximum sprout length following a culture period of 8 days. Capital letter pairs over plots indicate statistical comparison of corresponding data points. For all pair comparisons, *p* < 0.05, unless otherwise indicated. Error bars represent s.d. (*n* = 3).

**Figure 4 biomedicines-08-00365-f004:**
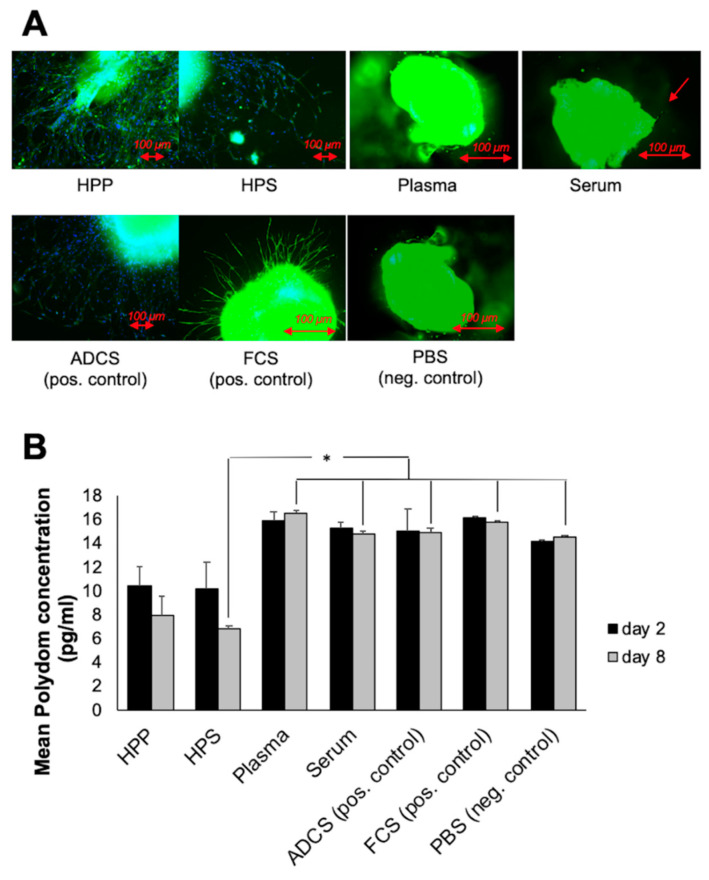
Validation of the lymphatic sprouting assay. (**A**) Panel showing representative images of immunohistochemical staining of thoracic duct rings with Anti-LYVE-1 (anti-lymphatic vessel endothelial hyaluronan receptor-1) (colored green) and DAPI (colored blue) in lymphatic sprouting assays (8 days), carried out in the presence of blood-derived secretomes (hypoxia preconditioned plasma (HPP) and serum (HPS), normal plasma and serum), as well as positive controls (adipose-derived cell suspension(ADCS) and fetal calf serum (FCS)) and negative control (phosphate buffered saline (PBS)) samples (Bars = 100 µm). (**B**) Quantitative analysis of polydom in thoracic duct ring assays. Plot showing the polydom concentration (pg/ml), measured in culture supernatants obtained following medium change in thoracic duct ring assays at 2 and 8 days. (* *p* < 0.05). Error bars represent s.d. (*n* = 3).

**Figure 5 biomedicines-08-00365-f005:**
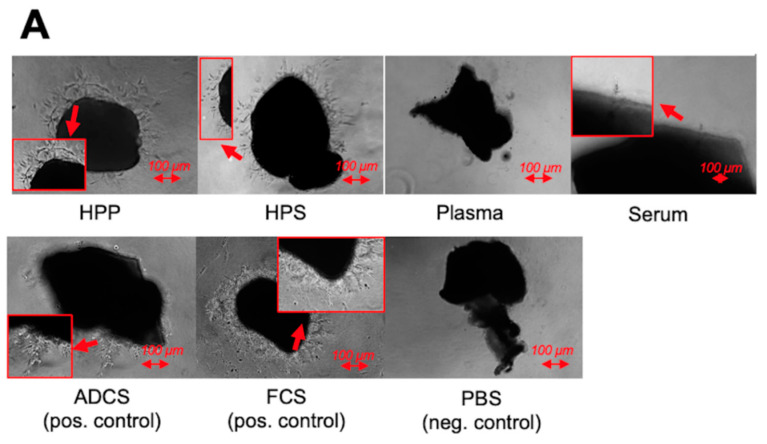
Effect of blood-derived secretomes on lymphatic sprouting in the thoracic duct ring assay. (**A**) Panel showing representative images of thoracic duct ring cultures (8 days culture), carried out in the presence of blood-derived secretomes (hypoxia preconditioned plasma (HPP) and serum (HPS), normal plasma and serum), as well as positive controls (adipose-derived cell suspension (ADCS) and fetal calf serum (FCS)), and negative control (phosphate buffered saline (PBS)) samples. Enlarged image sections are indicated by red insets. (Bars = 100 µm). (**B**) Plot showing the mean number of lymphatic sprouts formed in different culture conditions (2, 4, 6 and 8 days culture), as described above (*n* = 3). (**C**) Plot showing the mean maximum lymphatic sprout length (µm) after a culture period of 8 days (*n* = 3). Capital letter pairs over plots indicate statistical comparison of corresponding data points. For all pair comparisons, *p* < 0.05, unless otherwise indicated.

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
