# Peer review of "Effect of Hypoxia Preconditioned Secretomes on Lymphangiogenic and Angiogenic Sprouting: An in Vitro Analysis"

_biomedicines, 2020, doi:10.3390/biomedicines8090365_

Round 1

Reviewer 1 Report

 In the manuscript,  Moog and colleagues, using the assays of lymphatic sprouting from thoracic duct rings, show that HPS, and to a lesser extent, HPP are an autologous source of factors able of modulating lymphoangiogenesis (at least in vitro).

The manuscript is very well written, the data are well organized, convincing and well discussed.

Concerns

Although the authors describe very well the ADC isolation technique, they do not describe the protocol used for the collection of the conditioned medium, ACDS, of which they evaluate the pro-angiogenic and lympho-angiogenic efficacy (number of cultured cells, at which step after isolation, how long they condition the medium, in the presence or absence of serum...).

Why do the authors use female mice for aorta and thoracic duct isolation?

Why do the authors use 90 ng/ml VEGF as positive control? Is this concentration derived from pilot experiments?

What is the rationale for quantifying in conditioned media by the lymphatic rings, of the Polydom protein? The authors describe the results obtained convincingly, but it is not clear the rationale of such measurement.

Author Response

Dear Lacey Yuan, dear Reviewer,

thank you for the opportunity to improve our work and your valuable suggestions for improvement. We have worked hard to implement your suggestions.

We include with our revised submission an itemized, point-by-point response to the comments. Changes made in the manuscript have been highlighted.

We hope that you now consider our work to be suitable for publication in your valued journal.

Best regards,

Philipp Moog & Ulf Dornseifer & Ektoras Hadjipanayi

Point-by-point response:

Reviewer 1:

  1. Although the authors describe very well the ADC isolation technique, they do not describe the protocol used for the collection of the conditioned medium, ACDS, of which they evaluate the pro-angiogenic and lympho-angiogenic efficacy (number of cultured cells, at which step after isolation, how long they condition the medium, in the presence or absence of serum...).

Reply: We have taken the reviewer's comments into consideration, and have improved the corresponding part in M&M (section 2.3.) as follows:

“After this final isolation step, adipose-derived cells (ADCs) were resuspended in Dulbecco’s Modified Eagle’s Medium (D-MEM without serum, Life Technologies, Paisley, UK) to reach 3500 cells per mL. Cell counts were determined using the CASY Cell Counter & Analyzer (OLS OMNI Life Science GmbH & Co KG, Bremen, Germany) as described by the manufacturer’s protocol. Then adipose-derived cell suspension (ADCS) was ready for testing, without any further conditioning.”

After reconsideration, due to the lack of cultivation of adipose-derived cells, the nomenclature was appropriately adapted to "suspension", instead of ‘supernatant’. 

  1. Why do the authors use female mice for aorta and thoracic duct isolation?

Reply: The methodology of the aortic ring assay was investigated and published by Baker et al. in Nature Protocols in 2011. She describes that the sex of the mice does not appear to substantially influence aortic ring sprouting (Ref.: 36). Since the thoracic duct assay is an adaptation of  the aortic ring assay, this knowledge was adopted, by choosing to use female mice. Bruyere et al. also does not define gender in his protocol (Ref. 37). We therefore consistently used the same sex and age to avoid any bias.

Since the adipose tissue was removed from the same animals, these were also female mice.

  1. Why do the authors use 90 ng/ml VEGF as positive control? Is this concentration derived from pilot experiments?

Reply: The corresponding concentration was selected on the basis of unpublished preliminary tests, since no sprouting angiogenesis was found using lower concentrations. The concentration of 90ng / ml VEGF has already been used successfully as positive control in previous publications by our working group (Ref.: 23, 28, 29).

  1. What is the rationale for quantifying in conditioned media by the lymphatic rings, of the Polydom protein? The authors describe the results obtained convincingly, but it is not clear the rationale of such measurement.

Reply: Polydom measurements, as used here, were not a “test” of polydom protein concentration in conditioned media, but simply a confirmation of the lymphatic origin of sprouts. This complemented the immunohistochemical data. Further studies are required which will aid the understanding of the exact function of polydom in lymphangiogenesis, as highlighted in the discussion.

Reviewer 2 Report

In the present study the authors extended, however, partially repeated the findings of their previous work  (Biomedicines. 2020 Jan 16;8(1):16. doi: 10.3390/biomedicines8010016. PMID: 31963131).

Therefore, the novelty of the present study is limited. Novel is the part on the impact of hypoxia preconditioned plasma (HPP) and serum (HPS) on lymphangiogenesis.

The part of the present study related to the impact of HPP and HPS on angiogenesis (microvessel sprouting in the mouse aortic ring assay) is, as said a repetition: the results shown in Fig. 3 of the present study are very similar to those in Fig. 5 of the previous work. Similarly, the results shown in Fig. 2 in the present study (concentrations of growth factors in blood-derived secretoms) are similar to those in Fig. 2 of the previous work.

Additionally, Fig 1 should be combined with Fig. 4. Accordingly, 2 Figures remain in the manuscript: a new Fig. 4 (Fig. 1 + Fig. 4) and Fig. 5.

The authors should include additional controls in both angiogenesis and lymphangiogenesis assays:

  1. Normoxia preconditioned plasma (NPP) and serum (NPS) to assess whether 4 day-incubation of blood in the presence or absence of heparin under normoxic conditions affects the concnetrations of growth factors as well as angiogenic/lymphangiogenic - activity of plasma and serum compared to fresh plasma and serum as well as compared to HPP and HPS.
  2. The authors should quantify the amount of heparin which remains in the HPP preparation after 4 day-incubation and add the amount of heparin in to HPS to examine whether the higher capacity of HPS, compared to HPP, to induce lymphatic sprouting is mediated by residual heparin.

Minor.

legend to Fig. 5, lane 3: thotatic

Author Response

Dear Lacey Yuan, dear Reviewer,

thank you for the opportunity to improve our work and your valuable suggestions for improvement. We have worked hard to implement your suggestions.

We include with our revised submission an itemized, point-by-point response to the comments. Changes made in the manuscript have been highlighted.

We hope that you now consider our work to be suitable for publication in your valued journal.

Best regards,

Philipp Moog & Ulf Dornseifer & Ektoras Hadjipanayi

Reviewer 2:

  1. In the present study the authors extended, however, partially repeated the findings of their previous work  (Biomedicines. 2020 Jan 16;8(1):16. doi: 10.3390/biomedicines8010016. PMID: 31963131). Therefore, the novelty of the present study is limited. Novel is the part on the impact of hypoxia preconditioned plasma (HPP) and serum (HPS) on lymphangiogenesis. The part of the present study related to the impact of HPP and HPS on angiogenesis (microvessel sprouting in the mouse aortic ring assay) is, as said a repetition: the results shown in Fig. 3 of the present study are very similar to those in Fig. 5 of the previous work. Similarly, the results shown in Fig. 2 in the present study (concentrations of growth factors in blood-derived secretoms) are similar to those in Fig. 2 of the previous work.

Reply: The reviewer has raised a concern regarding the repetition of findings from our previous work. While we do agree that we had previously published our results relating to the angiogenic sprouting of HPP and HPS, it is important to note that these are autologous secretomes, whose composition, and subsequently their effect, varies widely from subject to subject. This can be indeed seen in the large standard deviation in our previous dataset. In the current study, we aimed to directly compare the angiogenic and lympangiogenic sprouting effect of HPP / HPS, which could only have been carried out by sampling these (fresh) secretomes from the same subjects. Furthermore, these secretomes were directly compared to a new positive control, ADCS, something that was not done in our previous study, which provides further insight. We therefore consider the fact that our new results are aligned with those of our previous work, a strength and validation of both studies, and as such provide a strong foundation for utilising hypoxia preconditioned secretomes as autologous therapeutic compositions.

The originality lies indeed in showing that both angiogenesis and lymphangiogenesis are substantially promoted by one approach. To our knowledge, this is one of the first studies (if not the first study) that directly compared angiogenic and lymphangiogenic sprouting side-by-side, using an (autologous) pharmaceutical composition derived from the same subjects.

  1. The authors should include additional controls in both angiogenesis and lymphangiogenesis assays:
  2. Normoxia preconditioned plasma (NPP) and serum (NPS) to assess whether 4 day-incubation of blood in the presence or absence of heparin under normoxic conditions affects the concnetrations of growth factors as well as angiogenic/lymphangiogenic - activity of plasma and serum compared to fresh plasma and serum as well as compared to HPP and HPS.
  3. The authors should quantify the amount of heparin which remains in the HPP preparation after 4 day-incubation and add the amount of heparin in to HPS to examine whether the higher capacity of HPS, compared to HPP, to induce lymphatic sprouting is mediated by residual heparin.

Reply: The aim of this study was to assess the ability of hypoxia preconditioned secretomes (HPP, HPS) to induce angiogenic and lymphangiogenic sprouting. These compositions, have already been thoroughly standardized and studied in various other assays, eg endothelial tube formation assay, fibroblast migration/proliferation assays etc. (see Refs 23,27). Therefore, this study aimed to provide further knowledge into the bioactivity of these secretomes, which are already in clinical use. While the proposed testing of normoxia preconditioned plama /serum (NPP, NPS), as suggested by the reviewer, is indeed interesting, it was beyond the scope of this study, as these compositions are not a true control, but rather new test compositions (of which currently little is known). Instead, here, we chose to use fresh plasma and serum, as controls, which have been previously used. It is important to note that the very process of HPP and HPS preparation utilizes a local (pericellular) cell-mediated hypoxia that is generated in situ under a normoxic environment, as described in methods section 2.2. In other words, no external hypoxic chamber was employed. We consider this as a key strength of the method, as is overcomes the need for expensive equipment. This, however, further complicates the distinction between HPP/HPS and NPP/NPS, since conditioning of peripheral blood within a normal syringe (blood volume per unit area >1ml/cm2) would naturally produce a pericellular hypoxic microenvironment. In order to prepare true NPP/NPS, the blood should be conditioned in larger containers, which would also change the blood cell seeing density (i.e. the secretomes would differ both in terms of oxygen tension and cell seeding density). This clearly is, therefore, a more complex question, which should be addressed in another, carefully designed study.

With regards to heparin, we had previously tested heparin in isolation, and found that it does not induce angiogenic sprouting (see Ref 28). However, heparin does not inhibit sprouting either, in contrast to other anticoagulants (e.g. EDTA). Based on these results, we have considered heparin to be a neutral agent in sprouting. The finding that HPP induced more lymphatic sprouting than fresh plasma, which also contained heparin, further highlights the fact that heparin is unlikely an inhibitor of sprouting. We have added a comment in discussion, for further clarification.

  1. Minor: legend to Fig. 5, lane 3: thotatic

Reply: Correction has been carried out, as requested.

Round 2

Reviewer 2 Report

The authors provided reasonable explanations and justifications regarding the reviewer's concerns.